# Refugee Status as a Possible Risk Factor for Childhood Enuresis

**DOI:** 10.3390/ijerph16071293

**Published:** 2019-04-11

**Authors:** Marija Jurković, Igor Tomašković, Mirna Tomašković, Branka Smital Zore, Ivan Pavić, Andrea Cvitković Roić

**Affiliations:** 1Faculty of Medicine, J. J. Strossmayer University, 31000 Osijek, Croatia; igor.tomaskovic@kbcsm.hr (I.T.); andreack@workmail.com (A.C.R.); 2Department of Pediatrics, Vinkovci General County Hospital, 32100 Vinkovci, Croatia; ipavic01@gmail.com; 3Department of Urology, Sisters of Mercy University Hospital Center, 10000 Zagreb, Croatia; 4Special Hospital for Protection of Children with Neurodevelopmental and Motor Disorders, Goljak, 10000 Zagreb, Croatia; brankasmital.zore@gmail.com; 5Zagreb Children’s Hospital, University of Zagreb School of Medicine, 10000 Zagreb, Croatia; ipavic01@gmail.com

**Keywords:** enuresis, child, prevalence, refugees

## Abstract

This study investigated the influence of refugee status on the occurrence of enuresis. It was performed among school children aged 6 to 11 years and their parents in the Vukovarsko-srijemska County (eastern Croatia), which had many displaced persons and refugees (mostly women and children) in the 1990s due to the wars in Croatia and Bosnia and Herzegovina. A specially designed questionnaire (about the child’s age and gender, the child’s enuresis history and that of the parents, and data on parental refugee status in childhood) was completed by one of the parents. Adequate data were collected for 3046 children. The prevalence of enuresis among the studied children was quite low (2.3%) but the prevalence distribution according to gender, the decline by age, and the higher odds ratio for paternal enuresis were in line with the results of other studies. The prevalence of parental enuresis in childhood was higher than their children’s enuresis (mothers: 5.8%, fathers: 3.6%, *p* < 0.001), and significantly higher among parents who had been refugees (mothers: *p* = 0.001, fathers: *p* = 0.04). Parental refugee status had no influence on the children’s enuresis. The results suggest that refugee status is a risk factor for the occurrence of enuresis in childhood.

## 1. Introduction

As defined by the International Children’s Continence Society, enuresis is any type of wetting episode that occurs in discrete amounts during sleep in children aged 5 or older after ruling out organic causes [1]. Almost to the end of the last century, nocturnal enuresis was considered to be a psychiatric/psychological disorder but the view has shifted towards somatic factors. Pathogenetically, most cases of nocturnal enuresis are the result of nocturnal polyuria or nocturnal detrusor over-activity, both combined with a high arousal threshold [2].

According to the current view, psychological problems may be a consequence, rather than the cause of enuresis, due to the distress and loss of self-esteem that are frequently reported [3]. This has been supported by studies that reported improvement in these problems after successful treatment of nocturnal enuresis [4,5]. On the other hand, there are studies where associations between psychological problems in early childhood and increased odds of nocturnal enuresis after the age of 5 were found [6]. It seems that environmental factors, including stressors in early childhood, could have an important role in its etiology. A possible mechanism is the suppressed release of antidiuretic hormones by the stress hormone cortisone which leads to polyuria [7].

Although some authors claim that psychological problems are overrated in the etiology of nocturnal enuresis, even they admit that psychological stress or trauma may be relevant, but only in a small subgroup of children with secondary nocturnal enuresis (relapse of enuresis after at least 6 months’ dry period) [2].

In earlier studies, most of the stressors examined were connected with family situations (the death or serious injury of a parent, financial problems, the birth of a sibling, or parental divorce or separation) [8]. More interest regarding the connection between enuresis and war stressors has arisen recently amongst refugees from war-torn areas in the Middle East and Asia, although this interest all originates from a single study group [9,10,11].

In the early 1990’s, Europe experienced a refugee crisis of an extent and scale not seen since the Second World War. Up until October 1992, approximately 2.5 million people from the former Yugoslavia were forced to leave their homes due to the threat of war or occupation. The Geneva Convention defines a refugee as a person who has a “well-founded fear of persecution for reasons of race, religion, nationality, membership of particular social group, or political opinion” (UNHCR 1951). In Croatia, a formal distinction was also made between displaced persons (individuals who were forced to leave their homes but remained within the borders of Croatia) and refugees (persons who fled their homes but crossed a state borders). Croatia took in the highest number of refugees in the region. In September 1992, there were almost 650,000 displaced persons and refugees in Croatia. The number of displaced and refugee children exceeded 100,000 [12].

The stress to which refugee children are exposed occurs in three stages: (1) while in the country of origin (witnessing violence, torture, loss of close family and friends, disruption of education, parental distress, general insecurity), (2) during the flight to safety (sometimes separated from parents), (3) when having to settle in a place of refuge—a period of “secondary trauma” (the need to settle into a new school, find a new peer group, and prematurely adopt adult roles) [13]. Refugee children experience the cumulative stress of forced migration, or the compounded stressors of childhood, combined with the traumatic experience of displacement [14].

Extreme violence of war torture affects children in different ways and may manifest immediately, physically or psychologically, or may remain hidden and unrecognized for years [15].

A broad spectrum of stress-related reactions was identified among displaced and refugee children (separation fears, withdrawal, aggression, sleeping and eating disorders) [16]. Numerous studies reported their levels of psychological distress [14] but specific data about the effect on enuresis are still scarce.

The aim of this study was to determine and compare the prevalence of enuresis among school children aged 6 to 11 years and their parents during childhood, and investigate if the status of displaced or refugee child is connected to the occurrence of enuresis.

## 2. Materials and Methods

A cross-sectional epidemiological study was performed from September 2018 to January 2019 by the completion of a special questionnaire by parents of school children aged 6 to 11 in elementary schools in the Vukovarsko-srijemska County. This county is situated in eastern Croatia and was severely affected by military activity in 1991 and 1992. In the middle of 1992, approximately 90,000 of its citizens, mostly women and children, were forced to leave their homes (nearly 39% of the total population in 1991). Up until 1994, the Vukovarsko-srijemska County also received about 25,000 refugees from Bosnia and Herzegovina [17]. When the war ended, most of the displaced persons returned to their homes. However, a vast number of refugees from Bosnia and Herzegovina settled in the area permanently. It may be expected that many of the parents of present-day school children were displaced persons or refugees and suffered wartime stress during their childhood.

Of 54 elementary schools in the county, 27 agreed to participate in the study, with 3921 school children aged 6 to 11, representing 65.5% of all children of that age in the county and 2.5% in Croatia.

The questionnaire was designed to investigate the characteristics of the child and the parents: the child’s age and gender, and the enuresis history of the child and the parents. Furthermore, data on parental status as a refugee or displaced person in childhood were collected. In this study both terms, refugee and displaced person, were used interchangeably. The purpose of the study was explained to the teachers who distributed one set of materials for each child: a copy of the questionnaire, a letter explaining the aims of the study, and a consent form. The mother or father or more rarely the legal guardian, completed the questionnaire and returned it to the teachers in the enclosed envelope. The person who answered the questions provided data about the particular school child and both parents.

The signed consent forms were collected separately from the questionnaires to protect privacy. The study was approved by the Medical Ethics Committee, Faculty of Medicine, Josip Juraj Strossmayer University, Osijek, Croatia (2158-61-07-19-01).

Incomplete questionnaires were excluded from the study.

The results are reported as frequencies and percentages. The χ^2^ test was used to study the associations between categorical variables. Odds ratios were calculated and a confidence interval (95% CI) was used. Differences were considered significant when *p* < 0.05. Statistical analyses were performed with MedCalc Statistical Software version 18.9 (MedCalc Software bvba, Ostend, Belgium; http://www.medcalc.org; 2018).

## 3. Results

Of 3921 distributed questionnaires, 3119 (79.5%) were completed, mostly by mothers (89.5%), then by fathers (9.9%) or other persons (0.5%). After exclusion of inadequate questionnaires, a total of 3046 questionnaires were analyzed. Most children (2224, 72.9%) were aged from 7 to 9 years (Figure 1).

The overall prevalence of enuresis among the studied children was 2.3% (71/3046). The prevalence of enuresis among boys and girls was 3.1% (47/1505) and 1.6% (24/1541), respectively (*p* = 0.004). The prevalence of enuresis according to age group declined from 3.4% at age 6 years to 1.7% at age 10 years (Table 1).

Data about the presence or absence of parental enuresis during their childhood were collected for 98.6% mothers (3003/3046) and 93.5% fathers (2850/3046). Of all the mothers and fathers who provided information on their own enuresis, 5.8% mothers and 3.6% fathers had had enuresis. A positive parental history of enuresis was found in 40.8% (29/71) of enuretic children. Odds ratios were calculated for the parent/child pairs with complete data sets. The odds ratios for enuresis in children increased if their parents also had had enuresis during childhood, and were 7.4 times higher for maternal and 9.5 times higher for paternal enuresis (Table 2).

Data about parental refugee or displaced person status in childhood were available for at least one parent for 99% of children (3017/3046); 98.4% for mothers (2968/3017) and 88.9% for fathers (2682/3017), respectively. In the sample of 5650 adults, 3349 (59.3%) of them were refugees or displaced persons in childhood. Significantly more mothers (6.9% vs. 3.9%, *p* = 0.001) and fathers (4.4% vs. 2.8%, *p* = 0.04) had enuresis if they were refugees or displaced persons during childhood. (Table 3).

Parental refugee or displaced person status in childhood did not lead to enuresis in their children (Table 4). 

## 4. Discussion

The aim of this study was to investigate the influence of refugee status on the occurrence of enuresis. After determining the major characteristics of enuresis among school children aged 6 to 11, we compared enuresis prevalence between the group of children and the group of their parents, and enuresis prevalence between the group of parents who were refugees in childhood and the group of parents who were not.

Studies that have investigated the influence of general war-related stress on enuresis in the former Yugoslavia are rare [18]. To the best of our knowledge, this is the first study to investigate the influence of displaced person or refugee status on enuresis among children exposed to the war in Croatia, or who were resettled in Croatia due to the war in Bosnia and Herzegovina in the 1990’s.

Recently, some studies have found a prevalence of enuresis of between 10.3 and 18.4% among refugee children resettled in Turkey. The sample sizes in these studies were small (38, 55 and 136 refugee children, respectively) due the fact that the data were collected in camps, camp schools, or health-care units [9,10,19].

Our research gathered information about childhood enuresis from a sample of adults who had been war-traumatized and forced to flee their homes during childhood. This is the largest studied group regarding this subject.

Different sample size and selection criteria in terms of age, and any bias during the informing process may cause differences in the reported prevalence of enuresis, but the most important determining factor is the definition of enuresis [20,21]. Although Neveus [2] stated that the prevalence of enuresis was similar in all cultures, the prevalence in the studies performed in Europe was 9–19% at the age of 5, 7–22% at the age of 7, 5–13% at the age of 9, and 1–2% at 16 years [22]. The overall prevalence of 2.3% among school children aged 6 to 11 determined in this study mostly supports the results of the only similar study undertaken in Croatia, where the prevalence was as low as 1.2%. The low rates of reported enuresis in our sample and in the sample studied by Miškulin et al. [23] may still be because of the social stigma of enuresis in Croatia.

A spontaneous enuresis resolution rate of 15% per year has been reported [24]. Despite the low prevalence, a similar decrease in the prevalence with increasing age was observed (Table 1).

The enuresis established in the study was significantly more frequent among boys than girls. This finding is consistent with many similar studies elsewhere [20,25,26,27,28], but there are some studies where girls had a higher prevalence [21,29,30,31].

The prevalence of maternal enuresis was higher than paternal, contrary to the prevalence of enuresis among their children, boys vs. girls. Whether the parental enuresis was primary or secondary was not investigated. Almost 60% of all the investigated parents were refugees in childhood and we may speculate that more cases of enuresis could have been secondary, caused by stress. It was shown that the girls were more stressed by refugee conditions than boys [10], so the higher prevalence of maternal enuresis is understandable.

As shown in Table 2, parental enuresis is significantly predictive of children’s enuresis with stronger association for fathers than mothers. Von Gontard at al. [32] found opposite associations but explained them by a possible recall bias.

A significant difference was found between the prevalence of enuresis in the examined children and their parents (Table 2) Significant differences in enuresis prevalence were also found between the groups of mothers and groups of fathers according to their refugee status in childhood (Table 3). Parental refugee status in childhood had no influence on the prevalence of enuresis in their children (Table 4). These results suggest that refugee/ displaced person status is a risk factor for the occurrence of enuresis in childhood.

The finding that the parental enuresis is predictive of enuresis in children is in line with the recent view that has shifted to somatic pathogenesis. However, the finding of parental refugee status predictive of childhood enuresis only in parents, not in their offspring, confirms that psychological factors also may have impact, irrespectively of inheritance.

The major strengths of the study are the large sample size and its epidemiological and representative design. This is the first study with a direct comparison of groups according to exposure to war related stressful events (children vs. parents; parents who were refugees vs. parents who were not refugees). This is also the first study about enuresis in refugee or displaced children in Croatia during the 1990’s. The limitations are that the results were possibly compromised by the potential retrospective recall biases common for this type of study. The parent who completed the questionnaire may not have been correctly informed about the other parent’s enuresis in childhood. Consequently, the prevalence of parental enuresis could be even higher and the difference between the two groups of parents who were refugees and those who were not could be even more significant. This research was performed in order to establish basic parameters that would lead to more comprehensive studies on the subject. Future studies should analyze other factors that could have an influence: psychosocial and demographic factors before the war, the precise experience of war, the age at the beginning and the duration of exile, and the place of resettlement. Information should be collected about the subtype of enuresis based on its onset (primary or secondary) and the age of onset and cessation of enuresis.

## 5. Conclusions

Forced migration is the fastest growing problem in society in the present-day world, causing multiple traumatic and posttraumatic experiences. Long-lasting and profound stress from ongoing conflict, violence, displacement, and the lack of basic life and psychosocial requirements lead to severe, often undetected symptomatology and disorders. Children who are refugees or displaced persons have an increased risk of the occurrence of enuresis, which can additionally traumatize them. Knowing about this risk of enuresis could help in providing adequate treatment.

The lower prevalence of enuresis in children compared to the enuresis prevalence of their parents who were refugees in childhood suggests the importance of environmental and stress factors in the etiology of enuresis.

## Figures and Tables

**Figure 1 ijerph-16-01293-f001:**
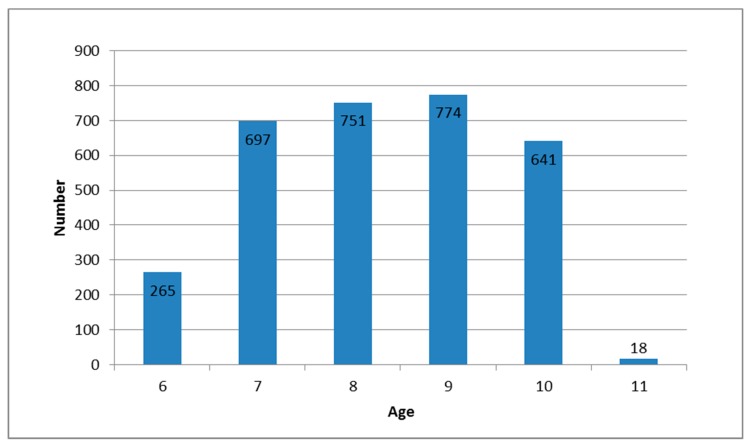
Age distribution of the investigated children.

**Table 1 ijerph-16-01293-t001:** Distribution of enuretic children according to age.

Age (Years)	Enuretic Children		Total
	*n*	%	*n*
6	9	3.4	256
7	16	2.3	697
8	18	2.4	751
9	19	2.5	774
10	9	1.7	541
11	0	0	18
Total	71	2.3	3046

**Table 2 ijerph-16-01293-t002:** Odds ratios for enuresis according to positive parental history of enuresis.

	Number of Children			Odds Ratios	95% CI	*p* *
	Enuretic	Non-Enuretic	Total			
Maternal enuresis						
Yes	20	153	173	7.4	4.3–12.8	<0.001
No	49	2781	2830			
Total	69	2934	3003			
Paternal enuresis						
Yes	15	89	104	9.5	5.1–17.6	<0.001
No	48	2698	2746			
Total	63	2787	2850			

* χ^2^ test.

**Table 3 ijerph-16-01293-t003:** Parental refugee or displaced person status and their enuresis in childhood.

	Parental Status of Refugee or Displaced Person			*p* *
	Yes	No	Total	
Maternal enuresis				
Yes	127	44	171	0.001
No	1709	1088	2797	
Total	1836	1132	2968	
Paternal enuresis				
Yes	66	33	99	0.04
No	1447	1136	2583	
Total	1513	1169	2682	

* χ^2^ test.

**Table 4 ijerph-16-01293-t004:** Parental refugee or displaced person status and their children’s enuresis.

	Number of Children			*p* *
	Enuretic	Non-Enuretic	Total	
Refugee or displaced person (mother or/and father)				
Yes	53	2066	2119	0.41
No	18	880	898	
Total	71	2946	3017	

* χ^2^ test.

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
