# Peer review of "Refugee Status as a Possible Risk Factor for Childhood Enuresis"

_ijerph, 2019, doi:10.3390/ijerph16071293_

Round 1

Reviewer 1 Report

Interesting article that falls in the interests of the journal.Clear structure, sound methodology, appropriate argumentation supported by  existing litterature.

 Appropriate writing tone.Minor syntax and grammar errors.

 Appropriate referencing system according to the journals standards

Author Response

Minor syntax and grammar errors were corrected by professional translator, native English speaker.

Reviewer 2 Report

The purpose of this study was to investigate refugee status on childhood enuresis, and concludes that refugee status is a risk factor for childhood enuresis. The study has several strengths: The first known study of its kind to investigate this phenomenon in displaced persons from this particular region, a large representative dataset, and both parental and offspring generations to tease out familial versus refugee status effects.  Certain aspects of the writing and presentation detract from the strengths of the paper, but they are relatively minor. I suggest the manuscript could be revised by considering the following:

1.       The authors’ conclusion based on their data findings should not be the title of the manuscript. The paper sought to investigate the whether or not refugee status is a risk factor, and the title should reflect that: E.g, “Refugee Status as a possible risk factor for Childhood Enuresis, “  or “The effects of Refugee Status on Childhood Enuresis.”

2.       The authors state in the introduction that the recent view has shifted to a somatic pathogenesis of enuresis rather than a psychological one.  How do the findings of parental refugee status predictive of childhood enuresis only in parents but not in children, AND the finding that parental enuresis status is predictive of enuresis in children, square off with either or both the psychological and somatic views? A paragraph in the discussion could drive this point more fully.

3.       The discussion section has frequent repetition of the statistics that are already (or should be) in the results section.  For instance, remove the p-values and percentages and Odds ratios that are already stated in the results.  E.g., the authors could reference results by stating something like this: “As shown in table 2, parental enuresis is significantly predictive of children’s enuresis, with a stronger association for fathers than mothers.”

Author Response

English language and style were corrected by professional translator, native English speaker. 

Point 1.       The authors’ conclusion based on their data findings should not be the title of the manuscript. The paper sought to investigate the whether or not refugee status is a risk factor, and the title should reflect that: E.g, “Refugee Status as a possible risk factor for Childhood Enuresis, “  or “The effects of Refugee Status on Childhood Enuresis.”

Response 1. We changed the title to emphasize the aim of the study: if refugee status could be a risk factor for childhood enuresis.

2.       The authors state in the introduction that the recent view has shifted to a somatic pathogenesis of enuresis rather than a psychological one.  How do the findings of parental refugee status predictive of childhood enuresis only in parents but not in children, AND the finding that parental enuresis status is predictive of enuresis in children, square off with either or both the psychological and somatic views? A paragraph in the discussion could drive this point more fully.

Response 2. One more paragraph was added to explain that our results could support possible "dual", not only somatic but also psychological view of pathogenesis of enuresis.

3.       The discussion section has frequent repetition of the statistics that are already (or should be) in the results section.  For instance, remove the p-values and percentages and Odds ratios that are already stated in the results.  E.g., the authors could reference results by stating something like this: “As shown in table 2, parental enuresis is significantly predictive of children’s enuresis, with a stronger association for fathers than mothers.”

Response 3. Some results were shifted to the results section. Repeated statistic data were removed. Numbers of tables were mentioned where considered to be  necessary.